



# Spectrometric fluorescence and Raman lidar: Absolute calibration of aerosol fluorescence spectra and fluorescence correction of humidity measurements

Jens Reichardt[1], Oliver Behrendt[1], and Felix Lauermann[1]

[1]Richard-Aßmann-Observatorium, Deutscher Wetterdienst, Lindenberg, Germany

**Correspondence:** Jens Reichardt (jens.reichardt@dwd.de)

**Abstract.** RAMSES is the operational spectrometric fluorescence and Raman lidar at the Lindenberg Meteorological Observatory. It employs three spectrometers, with the UVA spectrometer (378-458 nm spectral range) being the latest to be implemented in 2018. The UVA spectrometer extends the fluorescence measurement range to shorter wavelengths than previously accessible, and its water vapor measurements can be corrected for fluorescence effects. First the new experimental setup of the RAMSES

near-range receiver, which integrates the UVA spectrometer, is described. Then it is detailed how the fluorescence measurement with the UVA spectrometer is absolutely calibrated and how the fluorescence spectra are merged with those obtained with the second fluorescence spectrometer (440-750 nm spectral range). The second part of this study is dedicated to the effect of aerosol fluorescence on water vapor measurements with Raman lidars. When aerosols are present, a fluorescence-induced error always arises and therefore requires thorough analysis, even though it is particularly significant (in relative terms) only when

the atmosphere is dry, the fluorescence signal strong, or the bandwidth of the Raman detection channels wide. For moisture measurements with the UVA spectrometer, a method is introduced that effectively eliminates the systematic fluorescence error. However, the increase in trueness comes at the expense of precision. The investigations further show that an accurate correction for fluorescence is impossible when the Raman lidar is not equipped with a spectrometer but with a single fluorescence receiver channel only, at least for biomass burning aerosol, because for a given fluorescence backscatter coefficient at the wavelength

of the auxiliary detection channel the induced error in humidity can vary widely due to the changing shape of the fluorescence spectrum, which depends on aerosol type and atmospheric state and possibly also on other factors.

## 1  Introduction

Fluorescence lidars have long been an integral instrument in the environmental sciences. For example, they are of great importance in monitoring the condition of vegetation, buildings, and water bodies (e.g., Johansson et al., 1996; Raimondi et al.,

1998; Saito et al., 2016), or in hazard detection (e.g., Sato et al., 1978; Bobrovnikov and Gorlov, 2011; Li et al., 2019). In the study of atmospheric aerosol, on the other hand, fluorescence has long played a minor role compared to Raman scattering and has been treated almost stepmotherly, despite its potential to add a new dimension to the aerosol information space. Moreover, most instruments detect the fluorescence signal of aerosols only with a single broadband receiver channel (Rao et al., 2018; Veselovskii et al., 2020; Chouza et al., 2022), which means that the spectral properties must remain hidden. To the best of



the authors' knowledge, it was not until 2012 that the first lidar measurements of the fluorescence spectrum of atmospheric aerosols were published (Sugimoto et al., 2012), a breakthrough even though the calibration of the measurements leaves some questions. The first truly quantitative aerosol spectra were obtained somewhat later with RAMSES, the Raman lidar for atmospheric moisture sensing of the German Meteorological Service, in connection with spectral Raman measurements of clouds (Reichardt, 2014).

At the Meteorological Observatory Lindenberg, the RAMSES site, development of spectrometric lidar methods started in 2010. Initially designed as a conventional multi-parameter Raman lidar (Reichardt et al., 2012), RAMSES' measurement capability has been subsequently augmented by implementation of spectrometers serving different purposes. Starting with a subsystem specifically developed for the measurement of Raman spectra of clouds, the so-called water spectrometer (Reichardt, 2014), a second spectrometer (previously known as the fluorescence spectrometer, now referred to as the VIS spectrometer)

was added to the RAMSES far-range receiver which is exclusively dedicated to measurements of aerosol fluorescence at visible wavelengths > 440 nm. It is in routine operation since 2015. While rendering possible, for instance, studies of the co-existence of clouds and aerosols for the first time (Reichardt et al., 2018), its spectral restrictions allow only for a coarse aerosol typing and an error-prone absolute spectrum calibration. For these reasons it was decided to install a third spectrometer which would cover the UVA and blue wavelength range starting at about 378 nm. This new device, henceforth referred to as

the UVA spectrometer, had to be incorporated in the near-range receiver of RAMSES (requiring a thorough redesign), because in the far-range polychromator dichroic beamsplitters divert the UV wavelengths to the discrete lidar detection channels which forbids spectral analysis here. The UVA spectrometer was commissioned in 2018 and is operated routinely since 2020.

In this contribution the groundwork is laid for quantitative measurements of fluorescence spectra of atmospheric aerosols. Section 2 describes in detail the instrumental setup of the new near-range receiver and the UVA spectrometer. The VIS spec-

trometer and changes to the original design of the RAMSES transmitter are also discussed. Section 3 then provides a comprehensive account of the data processing chain that is required to obtain quantitative and complete fluorescence spectra. Section 4 investigates the effect of aerosol fluorescence on water vapor measurements with the Raman technique. The interference is known for a long time (Immler et al., 2005), and its gravity depends on the relative strengths of the water vapor Raman and the fluorescence signals. A method is devised for how humidity measurements with a spectrometer can be corrected for

aerosol fluorescence, and the performance of water vapor Raman lidars augmented by a single fluorescence receiver channel is assessed. Finally, Sect. 5 concludes this study and gives an outlook.

## 2 Instrument

Since relocation of RAMSES to the new lidar facility of the Lindenberg Meteorological Observatory (Reichardt et al., 2014), the instrument has undergone several upgrades, which are described below.



## 2.1 Transmitter

Early measurements with the VIS spectrometer showed that suppression of frequency-doubled Nd:YAG laser light was not high enough to avoid contamination of spectral fluorescence measurements in low clouds at 532 nm. The problem was solved by integrating a Pellin Broca prism into the transmitter setup and blocking the visible and infrared laser beams at the entrance aperture of the beam expander.

## 2.2 Far-range receiver

The VIS spectrometer was added to the RAMSES far-range receiver, it is in routine operation since 2015. In order to have all relevant spectrometric information at hand, some experimental details of the VIS spectrometer given by Reichardt et al. (2018) are summarized in the following. The setup consists of four main components, (1) a long-pass filter (Semrock RazorEdge, Rochester, New York) and the fiber coupler which images the atmospheric return signals transmitted by dichroic beamsplitter

BS7 (see Fig. 4b of Reichardt et al., 2012) onto the circular front face (2.3 mm diameter) of a, (2) quartz/quartz fiber bundle (100 μm cores, numerical aperture NA = 0.12). The fiber bundle, rectangularly reshaped, is coupled into a, (3) 1/8 m grating spectrograph (600 grooves/mm, 500 nm blaze). In its focal plane a, (4) 32-channel multianode photomultiplier single-photon-counting detection system is mounted (SP32-20; Licel GmbH, Berlin, Germany), yielding a spectral resolution of about 12.8 nm per detection channel. The wavelength range of the spectrometer is limited by the receiver optics preceding the edge filter

to the visible light spectrum between 440 and 750 nm.

## 2.3 Near-range receiver

For this reason, the UVA spectrometer had to be incorporated in the RAMSES near-range receiver which made its complete redesign necessary. On this occasion the original near-range Newtonian telescope was also replaced by a new and larger one (300 mm diameter, $f/4$; CT12 Orion Optics, Newcastle Under Lyme, Great Britain) which was recoated to enhance its UV

reflectivity to over 97 % (Tafelmaier Dünnschicht-Technik GmbH, Rosenheim, Germany). A fused-silica lens (50 mm focal length, both sides broadband anti-reflection coated like all other lenses in this setup) is used to reduce the effective focal length of the telescope slightly to $f_{\mathrm{eff}} = 1192$ mm (Fig. 1). The lens is part of the fiber coupler assembly which is mounted under a 45° angle with respect to the laser polarization plane to ensure the receiver is polarization-insensitive. The quartz fiber (800 μm core, NA = 0.22) that connects the telescope and the receiver is cut in two to accommodate an optic-fiber scrambler (3.18

mm sapphire ball lens; built to specification by Licel GmbH). This makes sure that range-dependent effects on the atmospheric profile associated with the lidar overlap function are avoided (Arshinov et al., 2004).

The atmospheric return signal enters the light-proof receiver housing via a near-UV achromatic lens ($f_{\mathrm{eff}} = 50$ mm; Edmund Optics, Barrington, NJ) for collimation. A quartz plate anti-reflection coated on both sides (Laser Components GmbH, Olching, Germany) is used to divert a tiny signal fraction towards the detector for elastically backscattered laser light (M), while the

main amount ($\sim 99$ %) is transmitted and passes through a sequence of optical elements [long-pass filter (Semrock) – notch filter (angle of incidence of 23°; Edmund Optics) – long-pass filter] that is designed to maximally suppress light around 355





nm in both the UVA spectrometer and the two discrete detection channels K and L for water vapor and molecular nitrogen Raman scattering at 407.5 and 386.7 nm, respectively. In addition, elastic and off-band background retroreflections are blocked with beam traps to minimize intra-receiver straylight, and particularly sensitive segments of the optical assembly are provided
with extra light-tight shielding.

By driving the linear stage, two operation modes can be chosen. Either the UVA spectrometer is illuminated directly, or the mounted dichroic beamsplitter deflects the light in the direction of the discrete detection channels. Usually, the former setting is used for measurements at night while the latter during daytime. This setup is similar to the one developed for the water spectrometer (Reichardt, 2014) where it has proven its flexibility in responding to different atmospheric conditions and
measurement tasks. Each discrete detection channel consists of a photomultiplier tube (R7400U/P-03; Hamamatsu Photonics, Hamamatsu, Japan) attached to a light-proof assembly containing a focusing lens system ($f_{eff} = 62.5$ mm), an interference filter (from the old near-range receiver; for center wavelength and bandwidth see Reichardt et al., 2012), and a box with five movable neutral-density filters for flexible signal-strength adjustment.

When the UVA spectrometer is activated, an achromatic doublet ($f_{eff} = 100$ mm; Edmund Optics) serves as a fiber coupler
which focuses the light on the front surface of a quartz/quartz fiber bundle (circular front-face aperture of 2.3 mm diameter, $NA = 0.12$). The trailing end of the bundle, now in rectangular shape, is plugged into an Oriel MS125-03 spectrograph (125 mm focal length, holographic grating with 2400 grooves/mm, 400 nm blaze) to which a 32-channel multianode photomultiplier row (SP32-200-HR, Licel GmbH) is attached. The detection system operates in photon-counting mode, temporal resolution and number of bins can be selected to optimize spectrometer performance. For routine operation, these are 50 ns
and 4000, respectively, and the spectral range is set to 378–458 nm.

Similarly to the water spectrometer (Reichardt et al., 2022), the UVA spectrometer was spectrally characterized at the ATLAS facility (Gröbner et al., 2016) of the World Radiation Center in Davos, Switzerland, in late 2017. Among other things, the linear dispersion was measured which is necessary for calculation of the spectral fluorescence backscatter coefficient. Detector bandwidth decreases with wavelength over the spectral range from 2.89 to 2.11 nm, at the $Q$ branch of the water
vapor Raman spectrum (407.5 nm) it is 2.64 nm.

Initial testing of the UVA spectrometer showed that signals from the vibrational-rotational Raman bands of $O_2$ and $N_2$ were so strong that they caused significant spill-over to neighboring detectors, thus compromising measurements. For this reason, a wavelength-dependent attenuator (Laseroptik GmbH, Garbsen, Germany) had to be inserted in front of the second edge filter which has a transmission of about 5 % at the respective $Q$ branches and $> 90$ % for wavelengths $> 400$ nm. The
test measurements also revealed that the elastically backscattered light caused considerable fluorescence in the fibers over the full spectral range. This detrimental effect was eliminated by covering the telescope with non-fluorescent acrylic glass, which attenuates the laser light at 355 nm by about 3 orders of magnitude (0.12 % transmission).





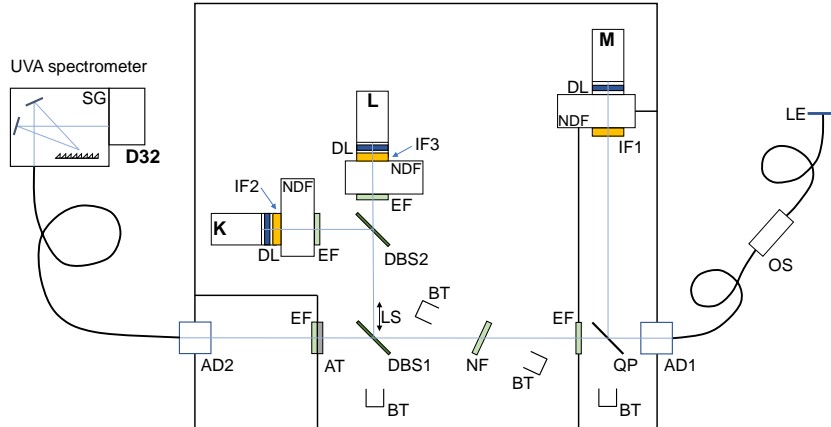

**Figure 1.** Schematic view of the RAMSES near-range receiver with three discrete detection channels (K–M) or, optionally, one discrete detection channel (M) and the UVA spectrometer. LE — lens, OS — optic-fiber scrambler, AD — achromatic doublets, QP — quartz plate, BT — beam traps, IF — interference filters, NDF — neutral-density filters, DL — double lenses, EF — edge filters, NF — notch filter, DBS — dichroic beamsplitters, LS — linear stage, AT — attenuator, SG — spectrograph, D32 — 32-channel detector.

## 3 Spectrum calibration and merging

### 3.1 Spectrum calibration

First a wavelength calibration of the fluorescence spectrometers was performed. In the case of the UVA spectrometer a mercury pencil lamp (LSP035; Quantum Design GmbH, Darmstadt, Germany) was installed above the near-range telescope close to the hatch. RAMSES was operated in standard mode, except that the atmospheric return signals were blocked at the entrance window of the telescope hall and an external delay generator was used to switch on the lamp in the time interval that corresponds to the altitude range from 19 and 29 km. For each of the strong Hg lines at 365.0, 404.7 and 435.8 nm (Sansonetti et al., 1996)

the micrometer screw of the grating drive was then tuned until two neighboring detectors showed similar strength of the height-integrated lamp signal, thus indicating the transition wavelength between both. This procedure was repeated with five detector pairs for each Hg line for a total of 15 micrometer-screw readings, which were input to a least-square-fit computer program yielding an optimum estimate of the center wavelengths of the 32 detector channels. For the VIS spectrometer a similar approach was chosen, only the light source was different. To provide for a better coverage of the spectrometer wavelength

range, a white-light light-emitting diode (LED) was employed and a narrowband interference filter installed at the entrance aperture of the far-range receiver for wavelength selection. Three interference filters with center wavelengths of 386.7, 532.1 and 660.5 nm were used successively.

Next, the relative intensity response of the spectrometers was determined. For the UVA spectrometer this was accomplished with spectrally calibrated UV LEDs positioned above the near-range telescope. LED calibration was performed at the in-house



Baseline Surface Radiation Network (BSRN) station. This approach was not viable in the case of the VIS spectrometer because the BSRN calibration setup exhibited a performance at longer wavelengths that is insufficient for this task. Instead, solar-background measurements in combination with radiative-transfer calculations were used. After extensive testing, the RAMSES data obtained in the morning of 9 October 2021 were selected for final analysis. A stack of additional, fully characterized neutral-density filters was inserted in front of the far-range receiver to protect the VIS spectrometer from the intense daylight.

For various solar zenith angles between $60°$ and $70°$, $180\,\mathrm{s}$ of spectrometer data were summed, integrated over all height bins and corrected for dark counts determined in measurements before and after the atmospheric observations. These measured spectra were compared to spectra obtained with the libRadtran (library for radiative transfer) software package (Emde et al., 2016) for the specific measurement configuration (zenith-pointing spectrometer with a field-of-view of $0.33\,\mathrm{mrad}$, photon-counting detection) and the prevailing atmospheric conditions. Concurrent spectral radiometer observations retrieved an aerosol

optical depth of 0.06 at $500\,\mathrm{nm}$, and extinction and backscatter Angstrom exponents of 1.18 and 0.025, respectively. A standard water vapor profile was assumed that was scaled to the measured integrated water vapor value of $5.8\,\mathrm{mm\,m^{-2}}$.

Figure 2 presents the results of the calibration effort. The spectra are normalized to their respective maxima. At shorter wavelengths sensitivity is low, either because of the optical attenuator implemented to weaken the signals of the $O_2$ and $N_2$ vibrational-rotational Raman bands (UVA spectrometer) or the dichroic beamsplitters required for the discrete detection chan-

nels (VIS spectrometer). The steady decline in relative sensitivity for longer wavelengths is caused by several factors of which the most important are decreasing grating performance and detector quantum yield. The sudden drop in UVA spectrometer sensitivity around $455\,\mathrm{nm}$ is due to a partial obstruction of the outermost detectors of the detector row.

Calibration of the spectrum in absolute terms is based on the water-vapor Raman technique, it follows closely the approach devised for the absolute calibration of the water spectrometer (Reichardt, 2014). However, as Fig. 2 illustrates, this method can

only be applied to the UVA spectrometer and not to the VIS spectrometer. Thus, to obtain a complete and absolutely calibrated fluorescence spectrum, the two partial spectra must be merged. This process is described in the following section.

## 3.2   Merging of UVA and VIS spectra

The relative-calibrated spectrum obtained with the VIS spectrometer is fitted to the absolute-calibrated spectrum measured with the UVA spectrometer over the wavelength range 441–456 nm. Here, and for longer wavelengths, the VIS measurement

is polarization-insensitive, and can therefore directly be compared with the UVA spectrum. A least-square fit is performed to obtain the proportionality factor between the two spectra, henceforth dubbed the calibration coefficient.

The method is demonstrated in the following taking the measurement of 21–22 August 2021 as an example (Fig. 3). During that night, aerosols were almost ubiquitous in the troposphere. Examining the elastic and fluorescence properties, one can distinguish between two different aerosol types. Below $3\,\mathrm{km}$, boundary-layer aerosol is present for which an extremely low

spectral fluorescence capacity (Reichardt, 2014, here its mean value in the 455–535 nm wavelength range is shown) is characteristic. In the free troposphere a significant biomass burning aerosol (BBA) event evolves over time. Aerosol scattering is so prominent that the elastic particle properties can be determined with high accuracy. As regularly observed, BBA fluorescence capacity is significant. Note that the elevated particle backscatter coefficients and depolarization ratios above $10\,\mathrm{km}$ and the





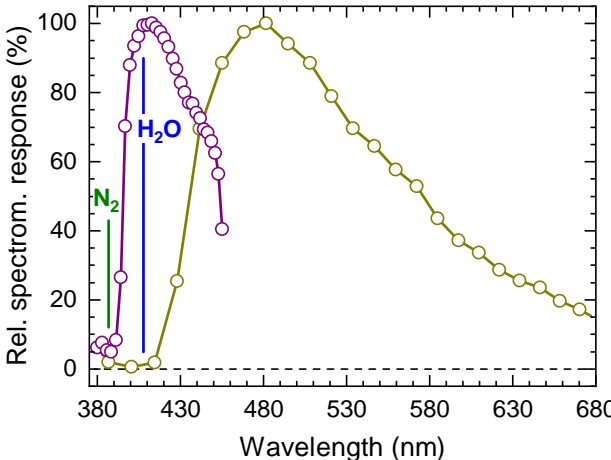

**Figure 2.** Relative calibration of the UVA spectrometer (violet curve) and the VIS spectrometer (dark yellow curve). Symbols indicate center wavelengths of the individual detectors. Raman $Q$ branch wavelengths of molecular nitrogen and water vapor are marked for reference.

low lidar ratios at the top of the boundary layer are features of cirrus and warm clouds, respectively, that were embedded in the
aerosol fields.

Figure 4 illustrates the merging of the UVA and VIS spectra for both the boundary-layer aerosol and the BBA. In the case of the former, Raman scattering by water vapor (and clouds, if present) below 415 nm and the vibrational overtone of $N_2$ Raman scattering at 424.4 nm (Knippers et al., 1985) overlap with the fluorescence spectrum. At higher altitudes or in the presence of strongly fluorescing aerosols, these contributions lose significance and eventually vanish (see Fig. 4b). The shape of the
fluorescence spectrum depends strongly on the aerosol type.

To have a set of parameters available to discuss the aerosol spectral characteristics, five fluorescence backscatter coefficients, or false colors, are defined as the sums of the spectral backscatter coefficient over certain wavelength intervals:

$$\mathcal{B}_{\mathrm{color}}^{\mathrm{FL}} = \sum_i \beta^{\mathrm{FL}}(\lambda_i)\Delta\lambda_i, \tag{1}$$

where $\beta^{\mathrm{FL}}$ is the spectral fluorescence backscatter coefficient measured at wavelength $\lambda_i$ with a detector bandwidth of $\Delta\lambda_i$.
The violet and blue fluorescence backscatter coefficients $\mathcal{B}_{\mathrm{violet}}^{\mathrm{FL}}$ and $\mathcal{B}_{\mathrm{blue}}^{\mathrm{FL}}$ cover the wavelength intervals 395–407 and 430–450 nm, respectively, and are obtained from UVA spectrometer data, the others ($\mathcal{B}_{\mathrm{cyan}}^{\mathrm{FL}}$, $\mathcal{B}_{\mathrm{green}}^{\mathrm{FL}}$, $\mathcal{B}_{\mathrm{red}}^{\mathrm{FL}}$) from VIS spectrometer data (wavelength ranges of 455–535, 545–620, and 630–710 nm, respectively). Further, the mean spectral fluorescence backscatter coefficient is defined as:

$$\beta_{\mathrm{color}}^{\mathrm{FL}} = \mathcal{B}_{\mathrm{color}}^{\mathrm{FL}} / \sum_i \Delta\lambda_i, \tag{2}$$

and the mean spectral fluorescence capacity (Reichardt, 2014) as:

$$C_{\mathrm{color}}^{\mathrm{FL}} = \beta_{\mathrm{color}}^{\mathrm{FL}} / \beta_{\mathrm{par}}, \tag{3}$$

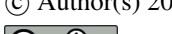



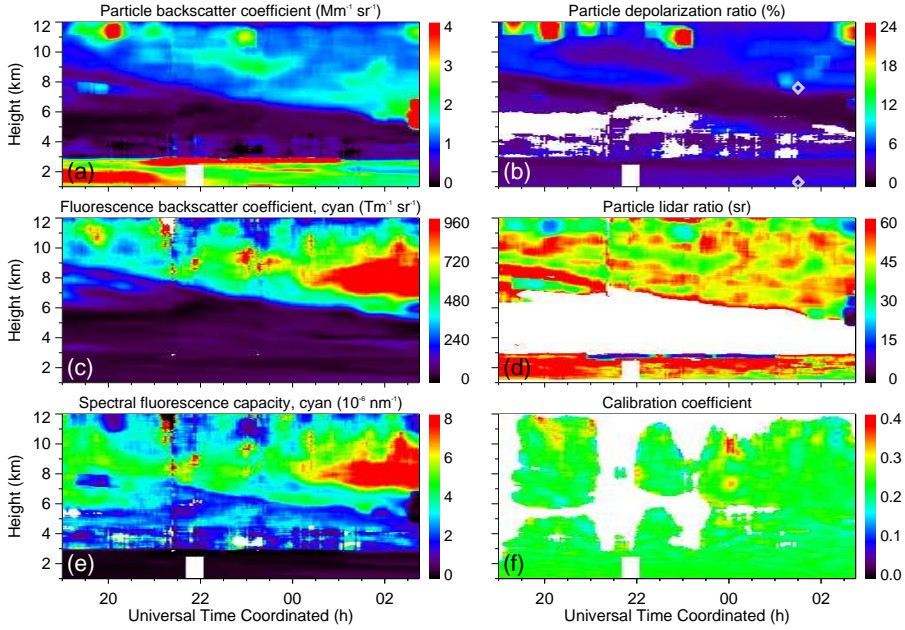

**Figure 3.** Temporal evolution of **(a)** particle backscatter coefficient, **(b)** particle depolarization ratio, **(c)** fluorescence backscatter coefficient (cyan false color: spectrum integrated from 455 to 535 nm), **(d)** particle lidar ratio (not corrected for multiple scattering), **(e)** spectral fluorescence capacity (mean value, 455–535 nm), and **(f)** calibration coefficient as measured with RAMSES in the night of 21–22 August 2021 between 19:00 and 02:50 UTC. The spectra measured at 1.3 and 7.6 km around 01:30 UTC [gray symbols, panel **(b)**] are analyzed in Fig. 4. For each profile, 1200 s of lidar data are integrated, the calculation step width is 120 s. The resolution of the raw data is 60 m, signal profiles are smoothed with a sliding-average length increasing with height. White areas indicate where data were rejected by the automated quality control process.

where $\beta_{\mathrm{par}}$ is the (elastic) particle backscatter coefficient. While $\beta_{\mathrm{color}}^{\mathrm{FL}}$ makes comparisons between measurements obtained with different fluorescence lidars possible, $C_{\mathrm{color}}^{\mathrm{FL}}$ is a measure of how strongly the observed aerosol fluoresces.

Even though the fluorescence spectra of boundary-layer aerosol and BBA particles greatly differ in shape and intensity, the
calibration coefficients deviate from one another by less than 5 % in this specific case. Extensive studies show that indeed the merging method is robust and universally applicable. Figure 3f provides an example of this finding. The contour plot of the calibration coefficient is remarkably homogeneous, independent of aerosol type, presence of clouds, and altitude. The distribution of the calibration coefficient is shown in Fig. 5. All nighttime data below 12 km are included, but filtered for elastic particle backscatter coefficients $> 10^{-7}\,\mathrm{m}^{-1}\,\mathrm{sr}^{-1}$ and particle backscatter ratios $< 3$ to limit statistical noise and exclude a
possible effect of clouds on the fluorescence spectrum, respectively. The distribution is relatively narrow (standard deviation of 0.02) and of Gaussian shape as the fit clearly shows, which indicates that the deviations from the mean are noise-related. The mean value of 0.22 can be regarded as a RAMSES-specific experimental constant, as it is typical of all fluorescence





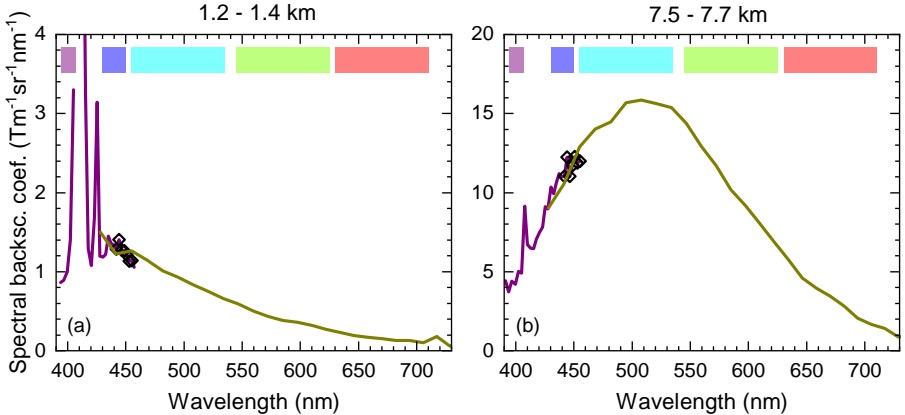

**Figure 4.** Determination of the calibration coefficient as exemplified by the RAMSES measurements at **(a)** 1.2–1.4 km and **(b)** 7.5–7.7 km on 22 August 2021, 01:30 UTC. The relative-calibrated spectra obtained with the VIS spectrometer (dark yellow curves) are fitted to the absolute-calibrated spectra of the UVA spectrometer (violet curves) over the wavelength range 441–456 nm (center wavelengths of UVA spectrometer detection channels indicated by black symbols), the calibration coefficients obtained are **(a)** 0.2213 and **(b)** 0.2312. Pastel violet, blue, cyan, green and red color bars show spectral integration ranges (395–407, 430–450, 455–535, 545–620, and 630–710 nm, respectively) of false color parameters of the aerosol fluorescence spectrum.

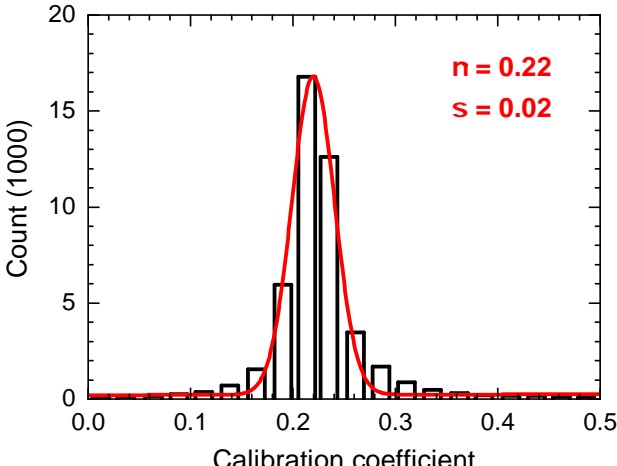

**Figure 5.** Determination of the calibration coefficient for the RAMSES measurements on 21–22 August 2021. Distribution of the calibration coefficient (bars) and its Gaussian fit (curve) are shown. $\mu$ — mean, $\sigma$ — standard deviation.

measurements investigated so far independently of month, season, or aerosol type. For this reason, it is used as the default value in the RAMSES data analysis.





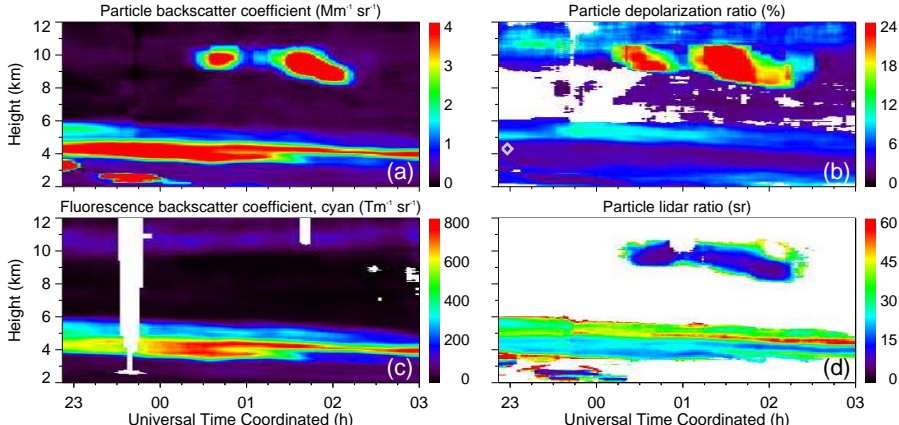

**Figure 6.** Temporal evolution of **(a)** particle backscatter coefficient, **(b)** particle depolarization ratio, **(c)** fluorescence backscatter coefficient (cyan false color), and **(d)** particle lidar ratio (not corrected for multiple scattering) as measured with RAMSES in the night of 9–10 August 2021 between 22:50 and 03:00 UTC. The spectrum measured at 4.3 km around 22:58 UTC [gray symbol, panel **(b)**] is analyzed in Fig. 7.

## 4  Fluorescence correction of humidity measurements

Figure 4 has illustrated that aerosol fluorescence can have an effect on water vapor measurements. It is to be expected that the magnitude of the induced error depends on the relative strengths of the water vapor Raman and the fluorescence signals, so it is the more severe the less humidity is available, the stronger the aerosols fluoresce and the wider the bandwidths of the detection channels. In Sect. 4.1 a case study is presented to illustrate the method that has been developed to correct water vapor measurements for fluorescence using the UVA spectrometer. Section 4.2 explores a more general correction technique which is based on a single discrete fluorescence detection channel only.

### 4.1  UVA spectrometer

Figure 6 displays the nighttime RAMSES observation of 9–10 August 2021 which is used to demonstrate the fluorescence correction method for humidity measurements with the UVA spectrometer. The date was chosen because of the clearly delineated and highly fluorescent aerosol layer between 3.8 and 6 km. The second layer at the tropopause and the partially overlapping cirrus cloud are intriguing features in their own right but are not discussed further. The data quality is generally very high, only around 23:40 UTC warm clouds at 2.5 km interfere with the measurement, leading to a rejection of the UVA spectrometer data above by the automated quality control due to unacceptable statistical errors.

Figure 7 highlights the calibrated UVA spectrometer data obtained at 4.3 km and 22:58 UTC (standard integration time of $\pm 600$ s). The local midnight radiosonde was launched at 22:48 UTC, so the RAMSES measurement spans the sonde flight from ground to the top of the lower aerosol layer at about 6 km (ascent speed of $\sim 5\,\mathrm{m\,s^{-1}}$). Below 394 nm vibrational-rotational $N_2$ Raman scattering governs the spectrum, above that wavelength the spectra of aerosol fluorescence, and water





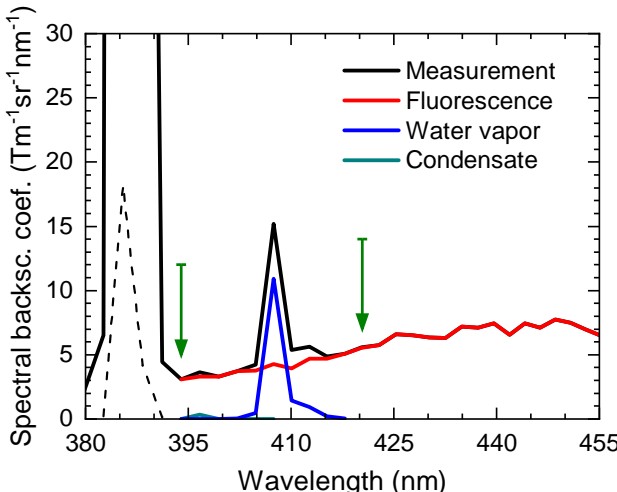

**Figure 7.** Decomposition of spectra obtained with the UVA spectrometer as exemplified by the RAMSES measurement at 4.3 km on 9 August 2021, 22:58 UTC. Center wavelengths of spectrometer detection channels that are unaffected by Raman scattering from molecular nitrogen, water vapor and condensate (liquid water, ice) are indicated (green arrows). The measurement is multiplied by a factor of 0.03 to fit into the display range around the vibrational-rotational Raman band of $N_2$ (dashed curve).

vapor and condensed water Raman scattering all add to the measurement, so spectral decomposition is required to separate the contributions. First, the spectral backscatter coefficients from the detection channels nearest to the water vapor $Q$ branch above

and below are selected, which are entirely dominated by fluorescence (see Fig. 7), similar to the approach taken by Liu et al. (2022). These data values are used to determine the spectral backscatter coefficient of fluorescence at 407.5 nm, the center of the vibrational-rotational water vapor Raman band. Linear interpolation is applied, any curvature in the fluorescence spectrum cannot be considered. However, visual analysis of the results suggests that the errors introduced by the linear approach should be small. Finally, the spectral backscatter coefficient of water vapor Raman scattering is obtained by subtraction, and converted

into the fluorescence-corrected water vapor mixing ratio using the calibration constant referred to in Sect. 3.1.

The spectrum decomposition is continued by first generating a synthetic water vapor Raman spectrum (blue curve, Fig. 7). It is the product of the spectral backscatter coefficient for water vapor Raman scattering at 407.5 nm just determined, and a normalized Raman reference spectrum. The reference spectrum is calculated according to Avila et al. (1999), and takes into account the actual ambient temperature and the spectral resolution of the spectrometer. Subtraction of the synthetic spectrum

removes the Raman scattering contribution of water vapor from the measurement. The remaining components (condensate and fluorescence) are separated from each other in a similar way as the water vapor Raman scattering from the total spectrum: the spectral backscatter coefficients of the detector channels encompassing the vibrational Raman spectra of liquid water and water ice are taken to determine the fluorescence spectrum within that wavelength range by linear interpolation.

The result of the spectral decomposition is shown in Fig. 7 also. The measurement splits into the fluorescence, water vapor,

and condensate spectra. The fluorescence spectrum is smooth and has no artificial features, which highlights the performance





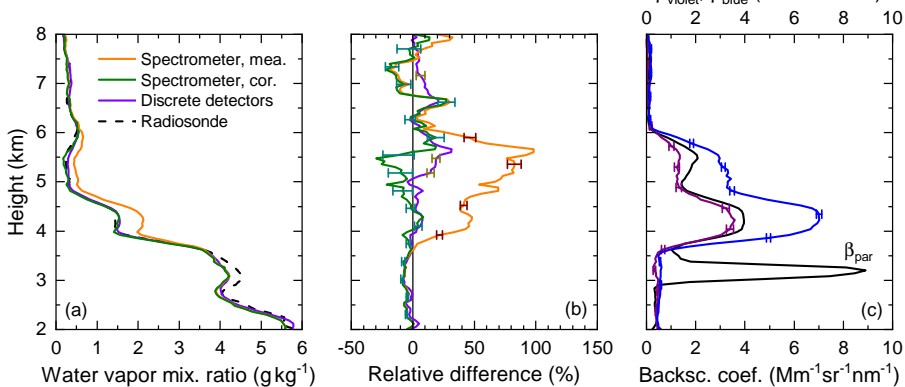

**Figure 8.** Effect of BBA fluorescence on water vapor mixing ratio measurements with RAMSES and its correction. Profiles of **(a)** water vapor mixing ratio as obtained with the UVA spectrometer (uncorrected and corrected for aerosol fluorescence) and discrete detectors (uncorrected), **(b)** relative difference between RAMSES and radiosonde water vapor mixing ratios, and **(c)** particle backscatter coefficient ($\beta_{\mathrm{par}}$) and mean violet and blue fluorescence spectral backscatter coefficients ($\beta_{\mathrm{violet}}^{\mathrm{FL}}$, $\beta_{\mathrm{blue}}^{\mathrm{FL}}$) measured on 9 August 2021, 22:58 UTC. The humidity profile of the routine midnight radiosonde launched on site at 22:48 UTC is shown for comparison [panel **(a)**]. Bars indicate statistical errors of the lidar measurement.

of the approach. However, since there are no clouds in this example, the small contribution of condensate is not genuine but an artifact of the spectral analysis caused by statistical noise. In general, the method is sensitive to noise when a significant fluorescence background is prevalent, which limits its scope, especially in the dry upper troposphere. The condensate spectrum may be employed to determine the water content of clouds, but only if the phase of the cloud is known. The spectral resolution of the UVA spectrometer is not sufficient to derive the fractions of liquid water and ice in mixed-phase clouds. Spectrometers with a spectral resolution $< 1\,\mathrm{nm}$ are required for this purpose (Reichardt, 2014).

Figure 8 presents the RAMSES profiles of 9 August 2021, 22:58 UTC, that are pertinent to this study. The violet and blue fluorescence false colors indicate the BBA layer between 3.6 and 6 km, the elastic particle backscatter coefficient follows the false colors closely here (Fig. 8c). Partly overlapping with the base of the BBA layer, a warm cloud is formed that extends to lower altitudes, but the elevated $\beta_{\mathrm{par}}$ values are not associated with higher $\beta^{\mathrm{FL}}$. During that night RAMSES was operated with the water spectrometer switched off, so water vapor mixing ratios are available from the UVA spectrometer (near-range receiver) and the discrete detection channels of the far-range receiver (Reichardt et al., 2012). The humidity profiles are shown in Fig. 8a, the concurrent radiosonde measurement is plotted for comparison. It is also used as a reference for the RAMSES measurements (Fig. 8b), notwithstanding that dynamics-induced differences in humidity may occur due to the drift of the probe. For instance, sensitivity studies confirm that the humidity differences between radiosonde and RAMSES around 3 km can be resolved if the launch date is slightly shifted which is an indicator for dynamical effects in the water vapor field.

Within the BBA layer, the uncorrected water vapor mixing ratios obtained with the UVA spectrometer exhibit a significant fluorescence-induced wet bias of up to 100 %. The magnitude of this wet bias is a function of fluorescence intensity and



available moisture. The fluorescence correction scheme discussed above removes effectively the wet bias, and the humidity
profiles of the UVA spectrometer and radiosonde agree in a statistical sense. Note, however, that the correction method leads
to higher statistical errors via error propagation. Thus, the trueness of the water vapor measurement is increased at the expense
of its precision. This fact may limit its usefulness in studies of BBA–cloud interaction in the upper troposphere.

An improvement in the moisture measurement can be achieved by using narrowband detectors because the contribution of
fluorescence to the apparent water vapor Raman signal is then less prominent. This is evidenced by the measurement with the
RAMSES far-range receiver, which uses interference filters of only 0.22 nm bandwidth in both the discrete water vapor and
$N_2$ Raman detection channels (Reichardt et al., 2012). Up to 5 km, the mixing ratio profile agrees well with the radiosounding,
even though it was not corrected for fluorescence interference (Figs. 8a and 8b). In the dry layer between 5 and 6 km, however,
fluorescence hampers the moisture measurement significantly, albeit much less than in the case of the UVA spectrometer.
Summarizing, no matter how good the performance parameters of a water vapor Raman lidar, fluorescence by atmospheric
aerosols may render the measurement useless without a correction. Chouza et al. (2022) arrived at the same conclusion in their
study of stratospheric humidity observations with high-performance Raman lidars. Section 4.2 investigates the prospects of
correction methods that are not based on a spectrometric approach but on the use of a single discrete fluorescence detection
channel.

Figure 9 visualizes the temporal development of the humidity field as measured with the RAMSES UVA spectrometer during
the night of 9–10 August 2021. Water vapor mixing ratio and relative humidity, both fluorescence-corrected, are shown, along
with the cyan fluorescence backscatter coefficient and the corresponding correction term for relative humidity. As expected,
the latter has the same spatiotemporal shape as the BBA layer, but shows differing internal patterns. For instance, the lower,
strongly fluorescing BBA filament observed around 4 km after midnight does not reveal itself as a clear feature of the wet
bias. The reason for this is that the shape of the fluorescence spectrum changes with altitude and thus, for the same cyan false
color, the spectral backscatter coefficient in the range of the water vapor Raman spectrum, which is, after all, decisive for the
correction, may well be different. This observation could already be made in Fig. 8, where the ratio of the two false colors
$\beta_{\mathrm{violet}}^{\mathrm{FL}}/\beta_{\mathrm{blue}}^{\mathrm{FL}}$ is about 0.51 at 4.2 km but only 0.39 around 5.2 km. Maximum relative errors in relative humidity are found in
the dry layer around 5 km where they can reach more than 100 %.

### 4.2 Single fluorescence detection channel

As a less costly and experimentally simpler means to study fluorescence of atmospheric aerosols than a spectrometer, the
implementation of a single discrete fluorescence receiver channel might be considered advantageous. For example, Veselovskii
et al. (2020) and Chouza et al. (2022) follow this approach to either investigate aerosols in their own right or to correct water
vapor measurements. To examine this conjecture, we assume in the following that RAMSES has only one discrete receiver
channel available to detect aerosol fluorescence. The blue false color was chosen because its spectral bandwidth (430–450 nm)
is similar to that selected byVeselovskii et al. (2020) (444–487 nm).

Figure 10 presents the relation between the mean blue fluorescence spectral backscatter coefficient and the induced error
in water vapor mixing ratio ($\Delta$MR) as measured on 9 August 2021, 22:58 UTC. The $\Delta$MR values reflect the differences





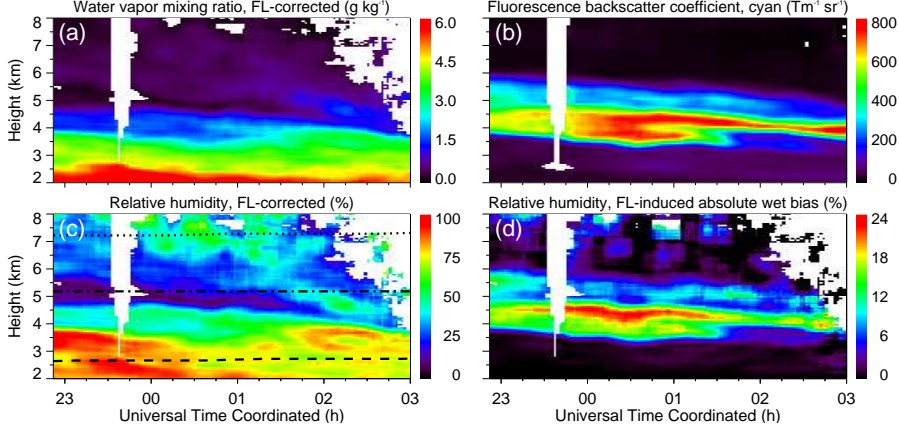

**Figure 9.** Measurement of **(a)** water vapor mixing ratio and **(c)** relative humidity (with respect to water and ice above and below $0\,^{\circ}$C, respectively; $0\,^{\circ}$C, $-15\,^{\circ}$C and $-30\,^{\circ}$C isotherms indicated by black lines) with the RAMSES UVA spectrometer in the night of 9–10 August 2021. The measurements are corrected for aerosol fluorescence. **(b)** Cyan fluorescence backscatter coefficient and **(d)** fluorescence-induced absolute wet bias of the spectrometer measurement (difference between measured and corrected relative humidity) are also shown.

between the uncorrected and the corrected UVA spectrometer measurements, the data points are taken from the altitude range between 3.6 and 6.1 km. As can be seen, $\Delta$MR and $\beta_{\mathrm{blue}}^{\mathrm{FL}}$ are well correlated. The relationship can be represented by a line

through the origin with gradient $m = 0.083\,\mathrm{m}^2\,\mathrm{sr}$. Its error is rather small, although single data points exhibit quite significant deviations. So at first sight it could be assumed that a correction of the water vapor measurement using a discrete fluorescence value is possible. On closer inspection, however, one notices that the data depicted in Fig. 10 show a small but systematic dependence on altitude, i.e. $\Delta$MR values from higher altitudes tend to lie above the compensation line and $\Delta$MR values from lower altitudes below it. In fact, this is not unexpected given the height dependence of the $\beta_{\mathrm{violet}}^{\mathrm{FL}}/\beta_{\mathrm{blue}}^{\mathrm{FL}}$ profile discussed earlier.

To further investigate the problem, other measurement nights were also analyzed. The results are summarized in Tab. 1. Moisture measurements with the water spectrometer or the discrete detection channels of RAMSES were examined for fluorescence effects as well. This task proved to be difficult because the fluorescence-induced measurement errors are much smaller because of the narrower detector bandwidths (0.79 and 0.22 nm, respectively). For this reason, BBA measurements in the dry upper troposphere had to be selected, for which the provision of a reliable reference profile was not a trivial task. The number

of measurement cases is therefore small. In the case of the water vapor measurements with the water spectrometer, the humidity field measured simultaneously with the UVA spectrometer served as a reference (corrected for fluorescence, acceptable statistical errors), and in the case of the RAMSES measurements with discrete Raman detection channels, the humidity profile of the local radiosonde (checked for drift-related systematic errors).

Figure 11 visualizes the fluorescence effect on water vapor measurements as a function of detector bandwidth. Two features stand out, the first is the large scatter of the gradient values at a detector bandwidth of 2.64 nm. As Fig. 12 will reveal,

this is related to the measurement situation and thus to the atmospheric state and to the type of BBA observed. This result





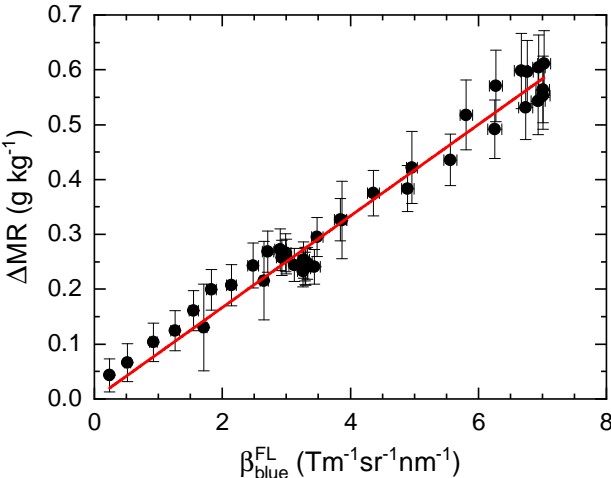

**Figure 10.** Relation between the magnitude of the water vapor mixing ratio correction ($\Delta$MR) and the mean blue fluorescence spectral backscatter coefficient ($\beta_{\mathrm{blue}}^{\mathrm{FL}}$) as obtained for the RAMSES measurement on 9 August 2021, 22:58 UTC. For the fit a straight line through the origin was assumed, and the entire altitude range of the BBA layer was considered (3.6–6.1 km). The fit yields for the gradient of the straight line $m = (0.083 \pm 0.001)$ m$^2$ sr, and for the correlation coefficient (squared) $r^2 = 0.994$.

clarifies that there cannot be a generalized correction of the fluorescence effect on the water vapor measurement; the specific fluorescence spectrum of each individual measurement must be taken into account. Second, the expected linear relationship exists between the magnitude of the fluorescence correction and the detector bandwidth. However, this only becomes evident

when measurements performed under similar conditions and on similar aerosol types are compared. For the data set discussed here, these are measurements at high altitudes and cold temperatures. Again, this observation can only be understood if one considers a dependence of the fluorescence spectrum on the state of the atmosphere.

Finally, Fig. 12 showcases the relationship between the characteristics of the fluorescence spectrum important for the correction of the water vapor measurement, the elastic optical particle properties of the BBA, and the measurement altitude and

temperature (aerosol layer integrated mean values). To the best of the authors' knowledge, this is the first time such a data set is presented. Obviously, there is a correlation between the different parameters, albeit a loose one. With gradient, false color ratio and ambient temperature decrease while elastic depolarization and lidar ratios increase. Phrased differently, BBA in the upper troposphere tends to have a fluorescence spectrum with higher backscatter coefficients at visible wavelengths and higher elastic depolarization and lidar ratios than at lower altitudes. The causes can be suspected at the molecular and microphysical

level, but certainly the source region and history of the BBA also play a role. Investigations on this topic are ongoing.





**Table 1.** Summary of humidity measurements in BBA layers.

| Subsystem FWHM (nm) | Date (yy/mm/dd) time (UTC) | Gradient (m² sr) | $r^2$ | Height (km) | Temperature (°C) | $\beta^{\mathrm{FL}}_{\mathrm{violet}}/\beta^{\mathrm{FL}}_{\mathrm{blue}}$ | $\delta_{\mathrm{par}}$ (%) | $S_{\mathrm{par}}$ (sr) |
|---|---|---|---|---|---|---|---|---|
| UVA spectrometer 2.64 | 21/07/18 22:48 | 0.069 | 0.998 | 3.0 | 3 | 0.48 | 3 | 30 |
| | 21/07/19 22:56 | 0.116 | 0.993 | 8.4 | -36 | 0.43 | 17 | 55 |
| | 21/07/30 22:54 | 0.088 | 0.988 | 4.9 | -9 | 0.52 | 6 | 33 |
| | 21/08/09 22:58 | 0.083 | 0.994 | 4.9 | -12 | 0.49 | 7 | 24 |
| | 21/08/21 22:45 | 0.100 | 0.986 | 7.2 | -23 | 0.41 | 2 | 44 |
| | 21/09/03 22:45 | 0.059 | 0.985 | 4.3 | -4 | 0.48 | 5 | – |
| | 21/09/29 22:45 | 0.152 | 0.989 | 10.8 | -52 | 0.39 | 22 | – |
| | 21/10/10 23:00 | 0.084 | 0.971 | 7.2 | -30 | 0.43 | 8 | 36 |
| Water spectrometer 0.79 | 21/08/22 01:00–02:20 | 0.043 | 0.904 | 5–10 | -25 | 0.50 | 5 | 50 |
| Discrete detectors 0.22 | 21/07/19 22:56 | 0.019 | 0.957 | 8.4 | -36 | 0.43 | 17 | 55 |
| | 21/09/29 22:45 | 0.017 | 0.970 | 10.6 | -52 | 0.41 | 22 | – |
| | 21/10/10 23:00 | 0.013 | 0.847 | 7.7 | -33 | 0.38 | 6 | 60 |

$r$ — correlation coefficient, $\beta^{\mathrm{FL}}$ — mean spectral fluorescence backscatter coefficient, $\delta_{\mathrm{par}}$ — particle depolarization ratio, $S_{\mathrm{par}}$ — particle lidar ratio

# 5 Summary and outlook

RAMSES is the operational spectrometric fluorescence and Raman lidar at the Lindenberg Meteorological Observatory of the German Meteorological Service. It employs three spectrometers, the water spectrometer (385–410 nm wavelength range), which is mainly dedicated to the determination of the water content in all three water phases (Reichardt, 2014; Reichardt et al., 2022), the UVA spectrometer (378–458 nm spectral range) for measurements of water vapor mixing ratio and aerosol fluorescence, and the VIS spectrometer (440–750 nm effective spectral range) for measurements of fluorescence at visible


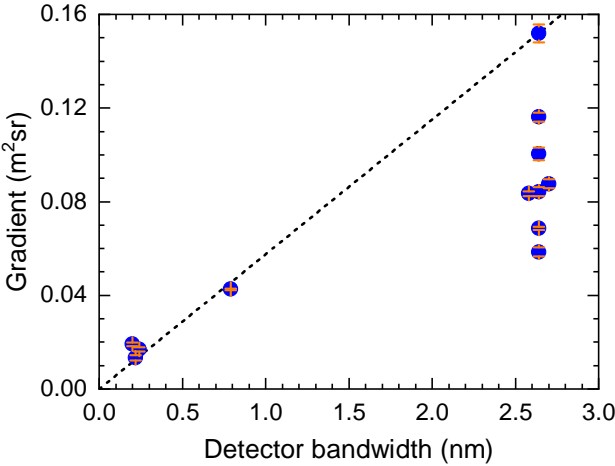

**Figure 11.** Impact of BBA fluorescence on the humidity measurement (expressed by gradient $m$) as a function of detector bandwidth. Results are shown for RAMSES measurements with the discrete detection channels (0.22 nm bandwidth), the water spectrometer (0.79 nm), and the UVA spectrometer (2.64 nm). Closely spaced data points were shifted slightly to separate them graphically. Error bars indicate retrieval uncertainties of $m$. The fit for the straight line through the origin was obtained by neglecting all values at 2.64 nm except for the maximum.

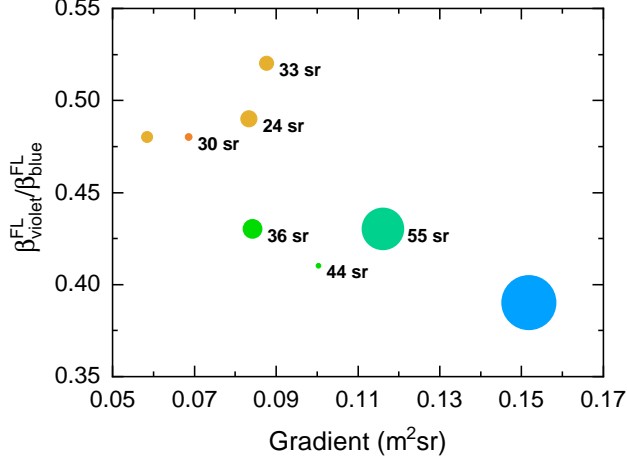

**Figure 12.** Ratio of mean violet and blue fluorescence spectral backscatter coefficients ($\beta_{\mathrm{violet}}^{\mathrm{FL}}/\beta_{\mathrm{blue}}^{\mathrm{FL}}$) as a function of gradient ($m$). Only measurements with the UVA spectrometer are considered. To visualize the relation to atmospheric temperature and particle optical properties, the data points are colored and sized according to ambient temperature and particle depolarization ratio, respectively, and particle lidar ratios are given as labels (when available). For numerical values, see Tab. 1.

wavelengths. In 2018, the UVA spectrometer was the latest subsystem to be implemented in the RAMSES instrument. While spectrometric studies were also previously possible with the VIS spectrometer alone (Reichardt et al., 2018), this addition greatly aided the absolute calibration of the fluorescence spectra. The UVA spectrometer also extends the fluorescence mea-





surement range to shorter wavelengths, and its water vapor measurements can be corrected for fluorescence effects. In the present publication, first the experimental setup of the near-range receiver, which had to be completely redesigned for the installation of the UVA spectrometer, has been thoroughly described. Then it has been explained in detail how the fluorescence measurement with the UVA spectrometer is absolutely calibrated and how the fluorescence spectra of the UVA and VIS spectrometers are merged.

The second part of this study has been dedicated to the effect of aerosol fluorescence on water vapor measurements with Raman lidars, first specifically for the UVA spectrometer, then in greater generality. When aerosols are present, a fluorescence-induced error always arises and therefore requires rigorous analysis, even though it is particularly significant (in relative terms) only when the atmosphere is dry, the fluorescence signal strong, or the bandwidth of the Raman detection channels wide. For moisture measurements performed with the UVA spectrometer, a method has been introduced that effectively eliminates

the systematic fluorescence error. However, the increase in trueness comes at the expense of precision, and this may limit its applicability to studies of aerosol-cloud interaction in the upper troposphere. It has then been investigated whether a correction of the profile would also be possible if no spectrometer but only a single discrete fluorescence receiver channel were available. For this purpose, the RAMSES measurements have been utilized to emulate the observations with such an instrument. The results do not support the hypothesis, at least in the case of BBA: For a given fluorescence backscatter coefficient at the

wavelength of the auxiliary detection channel the induced error in water vapor mixing ratio varies widely, which suggests that an accurate correction is impossible with this experimental approach. The underlying cause is the changing shape of the fluorescence spectrum, which certainly depends on the aerosol type and the atmospheric state, but possibly also on other factors such as cloud processing of the aerosols. Studies of this challenging scientific question are ongoing, and results will be published in a follow-up publication (Reichardt, J., et al.: Fluorescence spectra of airborne particles: dependence on source,

atmospheric state, elastic optical properties and clouds; in preparation).

A closing remark on water vapor measurements with Raman lidars in the upper troposphere and lower stratosphere: here the effect of BBA fluorescence cannot be ignored and requires correction. The results of this study suggest that a promising experimental approach would consider the use of a high-resolution spectrometer that includes the spectral range of the water vapor Raman band, or at least the use of two discrete fluorescence receiver channels, one at wavelengths immediately below

and one above the water vapor spectrum. The gradient of the fluorescence spectrum could be measured and thus the spectral fluorescence backscatter coefficient at the wavelength of the $Q$ branch required for the correction calculated. However, it remains to be seen whether the accuracy of the water vapor measurement that could be achieved in this way would be sufficient for studies of aerosol–cloud interaction or for monitoring of stratospheric humidity trend. Success will certainly depend on the overall performance of the Raman lidar, as already pointed out by Chouza et al. (2022) in their concluding comments.

*Data availability.* Lidar measurements are available upon request.





*Author contributions.*  JR developed the methodology, designed the RAMSES instrument, developed the analysis code and wrote the manuscript. OB helped perform lidar measurements and is in charge of the maintenance and operation of the lidar system. FL contributed to data analysis and helped with manuscript preparation.

*Competing interests.*  The authors declare that they have no conflict of interest.



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
