# Peer review of "Spectrometric fluorescence and Raman lidar: Absolute calibration of aerosol fluorescence spectra and fluorescence correction of humidity measurements"

_Atmospheric Measurement Techniques, 2022_

## Referee Comment (RC1)

Authors provide very detailed study of the fluorescence spectrum of aerosols (mainly BBA) and analyze the influence of fluorescence on water vapor measurements. This is high quality scientific study, which, in principle, can be published as it is.

I have just two comments.
- In this manuscript authors don't provide details of calculation of fluorescence backscattering. In particular, they don't mention how they performed the correction for aerosol differential extinction of fluorescence signals.
- 0.22 nm filter width in the far range water channel is rather small. Can temperature variations (and so variations of vapor spectrum) provide uncertainties?

---

## Author Response (AR1)

*The authors thank the three anonymous reviewers for their positive assessment of our manuscript and their valuable comments and suggestions.*

**REVIEWER COMMENTS**

**Reviewer #1: [posted on 22 Oct 2022]**

Authors provide very detailed study of the fluorescence spectrum of aerosols (mainly BBA) and analyze the influence of fluorescence on water vapor measurements. This is high quality scientific study, which, in principle, can be published as it is.

 I have just two comments.

- In this manuscript authors don't provide details of calculation of fluorescence backscattering. In particular, they don't mention how they performed the correction for aerosol differential extinction of fluorescence signals.
*Calculation/calibration of the fluorescence spectrum is performed analogously to that for the water Raman spectrum, including differential light extinction (Reichardt, 2014). Particle extinction is assumed to be wavelength-independent for clouds, and Angstrom coefficients derived from on-site spectral radiometer measurements are used for aerosols. The information is included in the revised manuscript at the end of Sect. 3.1.*

- 0.22 nm filter width in the far range water channel is rather small. Can temperature variations (and so variations of vapor spectrum) provide uncertainties?
*The lidar signals are corrected for the dependence of the Raman cross-section on ambient temperature, see Reichardt et al. (2012).*

**Reviewer #2: [posted on 05 Nov 2022]**

Summary

If the first part of this work, the authors describe the addition of a second spectrometer (and its calibration) to the RAMSES lidar. This new spectrometer covers the ~UVA range and is intended to provide additional information on aerosol fluorescence characteristics as well as a way to correct water vapor measurements for aerosol fluorescence contamination. In the second part of this work, the authors discuss the performance of this new correction approach and how it performs when compared to a single channel approach.

General comments

The paper is well written and provides many valuable experimental details with regard to the spectrometer calibration. It also highlights the importance of spectral measurements when looking at aerosol fluorescence and their potential use as an additional dimension in aerosol classification studies.

My main concern is related Sec. 4.2 "Single fluorescence detection channel", where the authors compare the spectrometer approach with the single channel method. The decision of the

authors to select the 430–450 nm band unnecessarily exacerbates the effect of the aerosol fluorescence spectral shape when compared to the solution presented in Chouza et al. (2022). Chouza et al. fluorescence monitoring channel sits at 410.3 nm, less than 3 nm apart from the center of the water vapor Raman spectrum, thus greatly reducing the impact of the changing shape of the aerosol fluorescence spectrum in the correction. It also needs to be mentioned that Veselovskii et al. (2022) used their aerosol fluorescence channel for aerosol research and not as a way to correct water vapor measurements, making the range selection less relevant.
*This fact is mentioned at the beginning of section 4.2.*

I suspect that the results of the spectrometer and single channel approach will be much more similar if the authors were to pick a much closer and narrower range. This is suggested by Fig. 7, where the difference in the fluorescence spectrum between 410 and 407 nm appears to be smaller than the random noise.
*The little wiggles are not noise-related but due to the imperfect correction for the water-vapor spectrum which stretches to wavenumbers > 4000 $cm^{-1}$.*

Generally speaking, while it is true that the spectrometer approach is potentially more accurate than a single channel correction approach, the much higher experimental complexity, potential calibration drifts and degraded precision should also be considered when comparing it with a combination of very narrow interference filter for the water vapor channel (0.22 nm as used in RAMSES far-range receiver) and a single fluorescence monitoring channel very close to the water vapor filter.
*According to the first author's referee comment on Chouza et al. (2022), a significant fluorescence effect on water vapor measurements is still to be expected even if the single fluorescence detection channel is only 3 nm apart from the center of the water vapor Raman spectrum. In the example presented, neglecting the gradient in the fluorescence spectrum (the spectral fluorescence backscatter coefficient increases by 7 % - 8 % from 407.5 to 410.5 nm) would yield a dry bias (https://doi.org/10.5194/amt-2022-98-RC2). Furthermore, setting a fluorescence channel too close on the long-wavelength side of the ro-vibrational water vapor Raman spectrum risks counting Raman signals as fluorescence, which would exacerbate the bias (Avila et al., 1999).*

*The authors believe that their discussion considers all the advantages and disadvantages of the different experimental approaches. The use of a single fluorescence channel spectrally close to a narrowband Raman channel, as described by Chouza et al. (2022), certainly represents a significant step toward fluorescence correction of water vapor measurements. However, this manuscript would like to suggest that even this approach may be improved, either by adding another discrete fluorescence channel (and optimizing the spectral positions of both), or by using a spectrometer. Whether the additional effort would be worthwhile, however, depends on the task at hand.*

Technical corrections

P13L285 a space is missing on "byVeselovskii".
*Corrected.*

**Reviewer #3: [posted on 17 Nov 2022]**

General comments:

The paper introduces the latest upgrade of the lidar system RAMSES and describes in great detail the procedures for setting up the instrument and characterizing it. Strong efforts were done to guarantee the quality of the measurements and products of RAMSES, which with its discrete and spectrometric channels, is the most advanced lidar in terms of fluorescence backscattering right now. The authors also evaluated the possibility of correcting the intrusion of fluorescence backscattering into water vapor channels, demonstrating that a correction is needed when fluorescence is strong and the ambiance is rather dry. The article is well-organized and well-written, and its relevance for the community can be quite high, given the rising demand for knowledge in terms of atmospheric fluorescence. There are some specific comments that I would like to be addressed by the authors.

Specific comments:

1) p.4, lines 91-93: What I get from the text is that (usually) during the night you let all signal go directly into the UVA spectrometer and during the day you habilitate the discrete channels by placing the beam splitter DBS1 on the way with the linear stage, as one can visualize in Fig. 1. I was wondering what is the reasoning behind that configuration? I would expect, e.g., the discrete channels to be off during the day because of the strong atmospheric background.
*This is correct, usually the three spectrometers are operated at night while during the day the discrete detection channels are utilized (see Lines 101-2). This is done for two reasons. First, the key measurement parameters of the spectrometers cannot be obtained in daylight because of the unfavorable signal-to-noise ratio. Second, the spectrometers have to be protected from the intense daylight. Note that the bandwidths of their row detectors are much wider than those of the discrete detection channels, so the latter are less affected by sunlight. Furthermore, signal levels in the discrete channels can be reduced by inserting additional neutral-density filters (Reichardt et al., 2012).*

2) P.5, Sec 3: As I could follow, the calibration has three steps: the wavelength calibration, the relative intensity response calibration, and finally the absolute intensity calibration. In the case of the VIS spectrometer, the second and third calibration procedures were not possible, so you proposed an alternative to these procedures. In order to do so, first, for the relative intensity calibration, you take the results from the libRadtran package. Could this process bring uncertainties to be considered by the approach? And regarding the absolute calibration, you scale the VIS relative-intensity spectra to the UV absolute-calibrated spectra. Can this scaling in the range between 441-456 nm carry uncertainties into the full calibrated spectra? What about extending this wavelength range?
*Possibly the largest uncertainty introduced by using the libRadtran package is due to the presence of aerosols because their wavelength-dependent light scattering properties have to be known. For this reason, we selected a day which was exceptionally clear, used data of co-located spectral radiometers to determine the (column-integrated) Angstrom coefficients of the aerosol present, and chose the measurements with the smallest solar zenith angle. Comparisons with measurements taken at other, larger solar zenith angles show excellent agreement, so we are confident that the aerosol effect was handled well.*
*Regarding the absolute calibration of the VIS spectrometer: Problems can arise when the UVA and/or VIS spectra are noisy which would result in a systematic offset of the VIS portion of the fluorescence spectrum. That is why we routinely check for each measurement night the*

*evolution of the calibration coefficient over time and height (Fig. 3) and its distribution (Fig. 5). All checks so far suggest that the calibration coefficient is stable and independent of aerosol type and atmospheric conditions when statistical noise is acceptable. For this reason, we apply this single, well-established value to all measurements, regardless of the noise level. Of course it would be nice to have a wider wavelength interval for determining the calibration coefficient, however, there are experimental limitations to this range: For shorter wavelengths the dichroic beamsplitters of the far-range receiver block light from entering the VIS spectrometer, for longer wavelengths the total spectral range of the UVA spectrometer would have to be moved which would impair measurements of the nitrogen Raman band.*

3) P.6, Line 135: What was exactly the issue with the relative intensity calibration at the BSRN station in the case of the VIS spectrometer? In general, I would assume there is more trouble when calibrating UV spectrometers. Could you provide a more precise comment on that?
*The problems with the BSRN spectrometer employed for calibrating the LEDs used for characterizing the RAMSES spectrometers were twofold. First, there was a software glitch which resulted in erroneous wavelength readings upon detector change-overs which made the VIS portion of the spectrum dubious. Second, the VIS detector was used up and thus not sensitive enough for our requirements. Both problems could not be resolved in time, so we had to come up with an alternative calibration approach for the VIS spectrometer.*

4) P. 6, Lines 166-167: You talk here about the high accuracy of the elastic particle properties. Could you provide more details about this processing? following your measurement strategy, it seems that during the night of 21-22 August, only the discrete channel 355 nm was on, but not the Raman channels. This means that you are using the Klett method to retrieve the aerosol properties. Or are you indeed using the Raman channels during the night, so that you can get the elastic backscattering by using the combination of Raman and elastic signals? Or are they from the far-range channels? Please provide information on where the information comes from. This would be helpful for the reader considering the complex lidar system you have.
*Actually, 'high accuracy' refers to the atmospheric conditions and the magnitude of the BBA event on this day which were quite exceptional (often it is easier to measure with RAMSES the fluorescence properties of aerosols rather than the elastic properties because of the extremely low aerosol extinction coefficients) and not to the retrieval method. With RAMSES, the elastic optical properties are always determined with the help of Raman detection channels, irrespectively of the receiver or operation of the spectrometers. The Klett method is not even implemented in the analysis software. Near-range data below 2-3 km and far-range data above are merged in case of duplicate measurement capability (e.g., water vapor mixing ratio, elastic particle properties), otherwise the profile is from one of the subsystems alone (e.g., pure-rotational Raman temperature, depolarization ratio, fluorescence spectra) and range restrictions may apply.*

5) P. 8, Fig. 3: similar to my last comment. When presenting the first results it seems there is missing information in the description of them, especially given the variety of receiver units and detectors you have. For example Fig. 3 a. Is this from the near-range or the far-range receiver? It s calculated from the Raman or Klett method? Did you have a fixed reference height for the whole period (for the high-resolution retrieval)? In Fig. 3b, I assume the depolarization is coming from the far-range channels, as you do not have them in the near-range receiver. For someone who read about the RAMSES for the first time, can be difficult to follow, considering that only a part of the system is described in this article. Most of the technical aspects can only be found in Reichardt et al 2012, which one can get only via institutional access. A more clear description in the text would be appreciated.

*It is always difficult to find a balance between repeating and citing previously published content. The authors lean to the latter option for this manuscript to keep its length in check and to help the readers' attention focus on the spectrometric aspects of RAMSES.* At the request of the reviewer, a brief description of the instrumental origin of the profiles in Fig. 3 has been included in its caption.

6) Figs. 3, 6, 8, 9: why do the products always start at 2 km height?
*In the latter part of this manuscript, the effect of BBA fluorescence on humidity measurements is discussed. Since it is more significant in drier air layers the focus is on the free troposphere. To show the boundary layer could unnecessarily distract the reader, and the relevant free-troposphere features would be visually less prominent. In Fig. 3, the height range actually starts lower at 1 km. Here this is warranted because we wanted to demonstrate the independence of the calibration coefficient under as many different atmospheric conditions as possible.*

7) 11, Line 225: The calibration constant (also named coefficient) was introduced in Sec. 3.2 instead of 3.1 as it is written.
*This is a misunderstanding because of poor wording. The reference is correct, it points to the constant determined by the absolute calibration of the spectrometer ('Calibration of the spectrum in absolute terms is based on the water-vapor Raman technique, it follows closely the approach devised for the absolute calibration of the water spectrometer (Reichardt, 2014).').* Wording has been changed.

8) 16, Table 1: What were the temporal windows considered on each individual measurement? +/- some minutes, or +/- tens of them?
*The center time of the integration period (+/- 600 s, s. caption Fig. 3) of each individual measurement is noted. The exception is the table entry for the water spectrometer, here not a single profile was analyzed but all profiles measured over the indicated time span.* This information is now included in the table.

9) 18, Lines 344-346: I am not sure I agree with the conclusion that an accurate correction is impossible. What do you mean by saying that *the induced error in water vapor mixing ratio varies widely*? You meant that the gradient varies widely? In that case, I would say it depends, on e.g., the bandwidth of the water vapor channel, at least for the discrete channels (0.22 nm bandwidth) the three gradient values you obtained (for the correction) are not so different from each other (0.013, 0.017, 0.019). In my opinion, a given value will provide a more accurate water vapor profile than a zero gradient (no correction). In general, what would your feasibility approach says if a discrete fluorescence channel is placed right next to the water vapor range, such as in Chouza et al., 2022, in which the fluorescence channel was intended for correcting the mentioned effect on the water vapor products?
*That is correct, the fluorescence error cannot be corrected accurately with a single-fluorescence-channel approach because the gradient of the fluorescence spectrum can vary. Certainly, correction with a zero gradient is better than no correction at all, and making an educated guess for the gradient is even superior, but a small uncertainty would remain. The authors answered the last question in their response to Reviewer #2, it is summarized here. As a measurement example demonstrates (https://doi.org/10.5194/amt-2022-98-RC2), the spectral fluorescence backscatter coefficient can increase by 7 % - 8 % from 407.5 to 410.5 nm. Thus neglecting the gradient in the fluorescence spectrum would result in a dry bias even if the fluorescence detection channel is only 3 nm apart from the water vapor Raman detection channel.*

*Further changes:*

- *Deleted reference to follow-up paper in Sect. 5.*